# Current Landscape of Immunotherapy for Advanced Sarcoma

**DOI:** 10.3390/cancers15082287

**Published:** 2023-04-13

**Authors:** Víctor Albarrán, María Luisa Villamayor, Javier Pozas, Jesús Chamorro, Diana Isabel Rosero, María San Román, Patricia Guerrero, Patricia Pérez de Aguado, Juan Carlos Calvo, Coral García de Quevedo, Carlos González, María Ángeles Vaz

**Affiliations:** Medical Oncology Department, Ramon y Cajal University Hospital, 28034 Madrid, Spain

**Keywords:** bone sarcoma, soft-tissue sarcoma, immunotherapy, checkpoint inhibitors, TCR, TIL, vaccines, adoptive cell therapy

## Abstract

**Simple Summary:**

The systemic treatment of advanced sarcoma remains challenging. Conventional chemotherapy and anti-angiogenic agents, even in the most responsive histologic subtypes, result in short responses and poor clinical outcomes. In a context where new therapeutic approaches are required, several strategies of immunotherapy have emerged as promising options, such as immune checkpoint inhibitors, vaccines, and adoptive cell therapy. In this review, we aim to summarize the current state and challenges of immunotherapy in patients with advanced bone and soft-tissue sarcomas.

**Abstract:**

There is substantial heterogeneity between different subtypes of sarcoma regarding their biological behavior and microenvironment, which impacts their responsiveness to immunotherapy. Alveolar soft-part sarcoma, synovial sarcoma and undifferentiated pleomorphic sarcoma show higher immunogenicity and better responses to checkpoint inhibitors. Combination strategies adding immunotherapy to chemotherapy and/or tyrosine–kinase inhibitors globally seem superior to single-agent schemes. Therapeutic vaccines and different forms of adoptive cell therapy, mainly engineered TCRs, CAR-T cells and TIL therapy, are emerging as new forms of immunotherapy for advanced solid tumors. Tumor lymphocytic infiltration and other prognostic and predictive biomarkers are under research.

## 1. Introduction

Sarcomas are a heterogeneous group of malignant tumors of mesenchymal origin. Their incidence in adults is low, comprising less than 1% of cancer diagnoses versus up to 15% of malignancies in the pediatric population [1]. More than 70 histologic subtypes have been identified, and they can be broadly classified into bone sarcomas (BS) and soft-tissue sarcomas (STS). The most frequent types of BS are osteosarcoma, chondrosarcoma (CS) and Ewing sarcoma (ES), whereas liposarcoma, leiomyosarcoma and undifferentiated pleomorphic sarcoma (UPS) are the most common subtypes of STS [2].

Conventional chemotherapy (CT) is still the standard treatment for unresectable or metastatic STS. Anthracyclines-based regimens, usually adriamycin plus ifosfamide, remain the upfront treatment [3], whereas other cytotoxic drugs are usually used in further lines (gemcitabine plus docetaxel [4], trabectedin [5,6], eribulin [7] or dacarbazine [8]). Several oral tyrosine–kinase inhibitors (TKI) have also demonstrated activity for STS, including multi-TKI pazopanib for non-adipocytic STS [9], anaplastic lymphoma kinase (ALK) inhibitors for myofibroblastic tumors with ALK fusions [10], and cediranib for alveolar soft part sarcoma (ASPS) [11].

In BS, multimodal treatment with CT, radiotherapy (RT) and radical surgery is recommended. For high-grade osteosarcoma, preoperative CT with a MAP regimen (doxorubicin, cisplatin, and high-dose methotrexate) is usually the front-line treatment for young patients [12]. In progressive disease, conventional CT with ifosfamide or cyclophosphamide plus carboplatin or etoposide is commonly used, with less evidence for other drugs, such as docetaxel and gemcitabine [13]. In ES, perioperative CT is indicated, usually with an interval VDC/IE scheme (vincristine, doxorubicin, cyclophosphamide, ifosfamide and etoposide) [14]. Topotecan plus cyclophosphamide and high-dose ifosfamide are the preferable options for further lines, followed by irinotecan plus temozolomide and docetaxel plus gemcitabine [15]. Several multi-TKI have also shown efficacy in advanced BS, mainly regorafenib, cabozantinib and apatinib for osteosarcoma [16].

Despite the recent incorporation of TKI and other drugs beyond conventional CT, the long-term prognosis of advanced sarcoma remains poor, with a median survival of 12–18 months for advanced STS [3], a 5-year survival rate <20% for osteosarcoma [17] and <40% for advanced ES [18]. The need for new therapeutic approaches, especially relevant given the predominance of these tumors in very young populations, explains the recurrent attempts to incorporate immunotherapy into the arsenal against advanced sarcoma.

The history of immunotherapy in sarcoma began with Coley’s inoculations of erysipelas, inducing tumor regression in some patients [19], though its development was at a standstill for many decades. High-dose interleukin-2 (IL-2) therapy, approved for advanced melanoma and renal cell carcinoma in 1998 [20,21], demonstrated some activity in pretreated pediatric sarcoma [22], though its use was restricted due to the high incidence of severe toxicity (cytokine-induced capillary leak syndrome [23]).

Modern immunotherapy with immune checkpoint inhibitors (ICI) has revolutionized the treatment of solid tumors. Though the results of ICI in monotherapy are poorer in sarcoma than in other malignancies, their combination with other agents seems to have synergistic effects, and promising strategies, such as vaccines and adoptive cell therapy, are emerging. However, there is a wide clinical heterogeneity between different histologic subtypes, disease settings and treatment categories [24]. This review aims to summarize the biological basis, current state, and future challenges of immunotherapy in advanced sarcoma.

## 2. Immunogenicity of Sarcoma: Anti-Tumor Response and Biomarkers

### 2.1. Innate Immunity and Release of Neoantigens

In sarcoma, as in other solid tumors, the activation and migration of cytotoxic T lymphocytes play a key role in the anti-tumor immune response [25] (Figure 1). Cancer cells are initially attacked by macrophages and natural killer (NK) cells, the main components of innate immunity, leading to cell death. Some ligands on the membrane of sarcoma cells activate NK cells, mainly through NK cell group 2D receptors (NKG2D) [26], and facilitate apoptosis. However, the essential mechanism of cell death is necrosis, with the subsequent release of tumor-associated antigens (TAA) with damage-associated molecular patterns (DAMPs), including aberrant ‘neoantigens’ produced as a result of accumulative somatic mutations [27].

Tumors with a higher mutational burden (TMB) have an increased level of neoantigens and a higher immunogenicity [28]. Sarcoma is a heterogeneous disease with significant variability in TMB among different subtypes, as demonstrated by Chalmers et al. [29]. The median TMB exceeds 20 mutations per Mb of DNA in angiosarcoma, leiomyosarcoma and UPS but is lower than two mutations per Mb in myxofibrosarcoma, liposarcoma and synovial sarcoma. However, the neoantigens burden related to the TMB is not the only factor that determines tumor immunogenicity. The presence of certain chromosomal translocations in tumor cells give rise to fusion proteins that bind to major histocompatibility complex class I (MHC-I) molecules and activate cytotoxic CD8+ T cells, working as powerful neoantigens [30]. Worley et al. [31] showed that some subtypes of sarcoma, such as clear cell, synovial and desmoplastic round cell tumors, can be highly immunogenic due to these genetic alterations, despite their low median TMB.

Kakimoto et al. [32] revealed that some high-grade sarcoma aberrantly express testis antigens (CTA), usually present in germ cells, such as melanoma-associated antigen (MAGE)-A4 and New York esophageal squamous cell carcinoma (NY-ESO)-1. MAGE-A4 was detected in 59% of synovial sarcoma and 56% of myxoid liposarcoma. NY-ESO-1 was found in 53% of synovial sarcomas.

The expression of NY-ESO-1 was associated with a better prognosis in high-grade sarcoma (5-year overall survival of 81% in the NY-ESO1+ group vs. 53% in NY-ESO-1- group, *p* < 0.05), presumably due to the powerful immunogenicity conferred by these fusion proteins.

Tumors with a higher mutational burden (TMB) have an increased level of neoantigens and a higher immunogenicity [28]. Sarcoma is a heterogeneous disease with significant variability in TMB among different subtypes, as demonstrated by Chalmers et al. [29]. The median TMB exceeds 20 mutations per Mb of DNA in angiosarcoma, leiomyosarcoma and UPS but is lower than two mutations per Mb in myxofibrosarcoma, liposarcoma and synovial. However, the neoantigens burden related to the TMB is not the only factor that determines tumor immunogenicity. The presence of certain chromosomal translocations in tumor cells gives rise to fusion proteins that bind to major histocompatibility complex class I (MHC-I) molecules and activate cytotoxic CD8+ T cells, working as powerful neoantigens [30]. Worley et al. [31] showed that some subtypes of sarcoma, such as clear cell, synovial and desmoplastic round cell tumors, can be highly immunogenic due to these genetic alterations, despite their low median TMB.

### 2.2. Antigenic Presentation and Activation of T Cells

After necrosis, dendritic cells phagocytose the released DAMPs and migrate to lymph nodes, where they work as antigen-presenting cells (APC). APC are essential for unleashing adaptive immunity by activating naïve CD8+ T lymphocytes (priming phase). This process needs the interaction of T-cell receptors (TCR) with MHC-I molecules on the APC surface, through which the phagocytosed antigens are presented [33].

T-cell activation also requires co-stimulatory signals, such as the coupling between co-receptor B7 (CD80) and the ligand CD28 on the APC surface. Certain cytokines released by APCs, mainly type I interferons (IFN) and interleukin-12 (IL-12), promote the activity of cytotoxic T cells and contribute to the activation of CD4+ helper T cells -following the coupling of TCR and MHC class II-. This facilitates B cell promotion and antibody production. Zhou et al. [34] showed that IL-12 up-regulates the expression of *Fas* receptors in osteosarcoma and ES cells, increasing their sensitivity to *Fas*-induced apoptosis. Type I IFN (IFN-α/β) have antiangiogenic and antiproliferative properties, which have been studied in models of Kaposi sarcoma [35] and angiosarcoma [36].

On the other side, some competitive co-receptors on the T cell membrane, such as CTLA4, LAG3 and TIM3, work as co-inhibitory signals that control this process in negative feedback [37]. Dancsok et al. [38] studied 1072 sarcoma specimens, revealing LAG3 and TIM3 expression on the infiltrating T-cells of nearly 50% of them, reaching 80% in some subtypes, dedifferentiated liposarcoma, myxofibrosarcoma and UPS. These ‘immune checkpoints’ are pathologically stimulated by tumor cells as a mechanism of immune escape, which sets the rationale for the use of ICI.

### 2.3. Tumor Infiltration of Activated Lymphocytes

Activated cytotoxic T lymphocytes reach the tumor via blood vessels and kill malignant cells in the peripheral tissues (effector phase). The successful trafficking of T cells to the tumor site is a key component of an effective immune response [25]. Once cytotoxic T cells are primed, they undergo a shift in the expression of surface proteins, losing CD62L and CCR7, which mediate their access to lymph nodes, and gaining molecules that facilitate their migration to diseased tissues. These include selectins, which facilitate the rolling of T cells on the endothelium, and receptors for inflammatory chemokines (CXCL9, CXCL10) that mediate their extravasation [39].

Tumor-infiltrating lymphocytes (TILs) comprise different subtypes of lymphocytes with high immunogenicity against tumor cells CD4+, CD8+, CD20+ and FoxP3+ TILs. CD8+ TILs interact through their TCR with antigens presented by MHC class I molecules on the surface of cancer cells, unleashing the cytotoxic cascade that leads to necrosis. In this phase, as in the priming phase, the anti-tumor response is controlled by immune checkpoints and can be suppressed by the activation of inhibitory co-receptors of the lymphocyte, such as programmed cell death receptor PD1, due to the interaction with immune-suppressive proteins expressed by tumor cells and cells from the tumor microenvironment (TME), such as PD1 ligand PDL1- [40].

D’Angelo et al. [41] analyzed the variability in PDL1 expression in tumors, lymphocytes and macrophages among different subtypes of sarcoma. The expression of PDL1 was more frequent in lymphocytes and macrophages than in the tumor cells, where it was detected just in three histologic subtypes, gastrointestinal stromal tumor (GIST), radiation-associated pleomorphic sarcoma and spindle cell sarcoma. Globally, the PDL1 expression was positive in six samples among 50 (12%) and was significantly associated with a high density of CD8+ TILs.

A meta-analysis by Zheng et al. [42], containing 15 studies and 1451 patients with bone and soft-tissue sarcoma, concluded that high expression levels of PDL1 were associated with poorer overall survival (HR 1.27, *p* < 0.001) and events-free survival (HR 2.05, *p* < 0.001), confirming the negative prognostic role of PDL1 expression. This is consistent with other studies; Que et al. [43] showed that a positive PDL1 expression is associated with Foxp3+ T-regs infiltration and a poor clinical prognosis in STS.

Whereas the ‘tumor killing’ role mainly corresponds to cytotoxic CD8+ TILs, CD4+ cells contribute to their priming and proliferation [44]. A higher CD4+ and CD8+ TILs infiltration is related to better prognosis in several solid tumors [45]. On the contrary, a high density of TILs expressing transcription factor forkhead box protein 3 (FoxP3), a marker of immune-suppressive regulatory T cells (T-regs), has shown a positive correlation with poor clinical prognosis [46,47].

Globally, sarcoma has lower TILs infiltration than other solid tumors, with huge heterogeneity among histological subtypes. D’Angelo et al. [41] also analyzed the percentage of TILs subsets (CD3+, CD4+, CD8+ and FOXP3+) among different subtypes of sarcoma. A ‘high density’, defined as >5%- of CD3+ cells, was frequently found in GIST (41%), angiosarcoma (14%) and spindle cell sarcoma (14%). A high density of CD4+ cells was present in GIST (50%), angiosarcoma (25%) and pleomorphic rhabdomyosarcoma (25%), whereas a high density of CD8+ cells prevailed in GIST (27%) and spindle cell sarcoma (18%). The prognostic impact of TILs in sarcoma is not fully clear, though some studies suggest better survival rates in patients with higher infiltration levels of CD4+ TILs [48] and CD8+ TILs [49].

Though T cells have been the focus of anti-tumor immunity research, B cells are progressively gaining strong attention. The development of B cells in the TME depends on the maturity of tertiary lymphoid structures (TLS). TLS are ectopic lymphoid organs developed in tissues under chronic inflammation, including tumors. In immature TLS, B cells evolve as T-regs and release immune-suppressive cytokines, whereas, in mature TLS with a germinal center, B cells undergo affinity maturation and isotypic switching, resulting in plasmatic cells that secrete anti-tumor antibodies [50]. As a favorable lymphocytic infiltration, the presence of mature TLS has been associated with better clinical outcomes, and strategies to induce TLS neogenesis in immune-low tumors represent a promising pathway for cancer immunotherapy [51].

A favorable B population seems to have a key role in the response against sarcoma. Sorbye et al. [52] showed that a higher density of CD20+ TILs in STS is an independent positive prognostic factor. Petitprez et al. [53] proposed an immune-based classification of STS based on the TME composition, identifying an immune-high class E (SIC E) whose specimens were particularly rich in CD20+ TILs. They analyzed 47 patients with STS from the SARC028 trial [54] and found that a high infiltration of CD20+ cells determined the highest response rate and progression-free survival (PFS) to PD1 blockade, even in tumors with low CD8+ TILs infiltration.

### 2.4. Immune-Suppressive Tumor Microenvironment

The TME dynamics are affected by complex reciprocal interactions between immune-stimulatory and immune-suppressive cells. The recognition of tumor immunogenic epitopes by TILs promotes tumor regression by activating tumors into a T-cell-inflamed ‘hot’ state. On the contrary, immune-suppressive cells such as T-regs, myeloid-derived suppressor cells (MDSCs) and tumor-associated macrophages (TAMs) interfere with effector T cells and facilitate tumor immune escape [55]. The predominance of these immune-suppressive cells leads to non-T-cell inflamed or ‘cold’ tumors, promoting tumor progression and impoverishing clinical prognosis.

T-regs suppress the function of effector cells by direct contact through the interaction between granzymes and perforins with the CD8+ T cell membrane [56] but also by indirect mechanisms, such as the release of inhibitory cytokines growth factor β (TGF-β), interleukin-10 (IL-10), IL-35 and prostaglandin E2 (PGE2) [55] and the suppression of APCs through the downregulation of CD80 and other stimulatory coreceptors [57]. Smolle et al. [58] analyzed the infiltration of CD3+ FoxP3+ T-regs in 192 surgical samples of STS and found an increased risk of local recurrence in tumors with CD3+ FoxP3+ T-regs. An increasing prevalence of T cells with a regulatory phenotype (CD4+, FoxP3+) has been found in the advanced stages of Ewing [59] and Kaposi sarcoma [60].

MDSCs facilitate epithelial-to-mesenchymal transition [61], act as mediators of neo-angiogenesis through the release of vascular endothelial growth factor (VEGF), fibroblast growth factors (FGF) and matrix metalloproteinase 9 (MMP9) [62] and induce TME remodeling by establishing a ‘pre-metastatic niche’ [63]. The MDSCs trafficking to the tumor is mainly mediated by the CXCR2 receptor, which has been proposed as a potential target to alter the TME and attenuate tumor progression [64]. Sarcoma cells can produce CXCR2 ligands, such as CXCL8, that facilitate the arrival of MDSCs to the TME. Highfill et al. [65] showed that pediatric patients with advanced sarcoma display elevated serum levels of CXCL8, which are associated with poor survival rates. In murine models, the blockade of CXCR2 seems to suppress MDSCs trafficking and enhance the anti-tumor activity of PD1 blockade. These findings suggest that strategies to prevent the trafficking of MDSCs to the tumor bed may improve the efficacy of checkpoint inhibitors.

Monocyte-related MDSCs (M-MDSCs), together with circulating monocytes and tissue-resident macrophages, after being recruited to the tumor site in response to colony-stimulating factors (CSF) and several chemokines, can differentiate into TAMs [66]. TAMs can be polarized to M1-like (classically activated) or M2-like (alternatively activated) macrophages. TAMs with a M1-like phenotype display anti-tumor functions, whereas the M2-like phenotype is associated with pro-tumorigenic activity [67]. In fact, a high density of M2-like TAMs in the TME has been associated with poor clinical outcomes in many solid tumors [68].

The unfavorable prognostic role of M2-like TAMs has been established in STS. Higher levels of TAMs expressing M2-related markers (CD163+/CD204+) have been associated with poorer survival and higher disease stage in leiomyosarcoma [69], myxoid liposarcoma [70], synovial sarcoma [71] and UPS [72]. The prognostic significance of the M1/M2-phenotype in bone sarcoma is more controversial [73]. Some studies suggest a positive impact of polarized macrophages with an M1 phenotype [74], but others have found no clear correlation with survival [75]. Some studies have even reported longer survival rates in osteosarcoma [76] and ES [77] patients with a high density of M2-like TAMs.

Several cytokines released by malignant cells promote the production of indoleamine 2,3-dioxygenase (IDO1) and VEGF. IDO1 is an intracellular enzyme that reduces the activity of effector T cells through the suppression of the tryptophan pathway [25]. Some studies have suggested that the IDO1 pathway could contribute to the immune-suppressive phenotype of sarcoma cells and be a relevant mechanism of their primary resistance to PD1 blockade [78]. In fact, a high IDO1 expression may be used as a biomarker of poor response to anti-PD1 agents in sarcoma [79]. Around 39% of human sarcoma express IDO1, especially when the CD8+ TILs infiltration is high, setting a rationale for the dual blockade of IDO1 and immune checkpoints [80].

VEGF interferes with an antigenic presentation by inhibiting the maturation of dendritic cells [81], restricts the migration of lymphocytes into the tumor compartment [82], and favors the recruitment of T-regs, MDSCs and TAMs, contributing to a highly immune-suppressive TME [83]. The growth and dissemination of sarcoma strongly depend on angiogenesis, and VEGF circulating levels correlate with stage, grade, and risk of metastasis [84]. Overcoming these barriers of the TME remains a major challenge to move immunotherapy forward in advanced sarcoma.

## 3. Immunotherapy for Sarcoma: Clinical Results

### 3.1. Immune Checkpoint Inhibitors (ICIs)

The first studies with single-agent immunotherapy failed to demonstrate a significant anti-tumor activity (see Table 1). Anti-CTLA4 antibody ipilimumab in monotherapy showed negative results in recurrent synovial sarcoma [85] and pediatric sarcoma [86]. Anti-PD1 nivolumab was tried in the third line in 12 patients with advanced uterine leiomyosarcoma, with no objective responses [87].

The first immunotherapy trial with positive results in sarcoma was phase II SARC028 with anti-PD1 pembrolizumab, including a cohort for BS and a cohort for STS [54]. In the BS cohort, there were only two objective responses (one osteosarcoma and one CS). In the STS cohort, with a total of 40 patients, there were seven responses (17.5%): four in UPS (including one complete response), two in dedifferentiated liposarcoma (DDLPS) and one in synovial sarcoma. Two expansion cohorts in UPS and DDLPS reported an objective response rate (ORR) of 23% and 10%, respectively [124]. Keung et al. [125] showed that patients from SARC028 who responded to pembrolizumab had higher densities of activated CD8+ TILs and an increased pre-treatment percentage of TAMs expressing PDL1 compared to non-responders.

Liu et al. [126] confirmed the activity of pembrolizumab in advanced STS in a real-world study, reporting an overall ORR of 19.4%.

Following the results of SARC028, a phase II randomized trial is currently studying neoadjuvant pembrolizumab combined with RT in high-risk UPS or DDLPS of the extremities (SUC2C-SARC032) [127]. Another phase II trial (STEREOSARC) is evaluating concomitant RT with atezolizumab in oligometastatic STS [128]. There is a rationale for combining ICIs and RT since RT induces the release of TAA following immunogenic cell death, which activates TILs and leads to the recruitment of more effector cells to the TME [129]. The primary endpoint is the progression-free survival rate at 6 months. If the results of these studies are favorable, ICIs may also gain ground in the context of early-stage disease.

ICIs have also been tested in less common histologic subtypes. The phase II trial AcSé evaluates pembrolizumab in different cohorts of patients with rare cancers. A total of 98 patients were enrolled in the sarcoma cohort [88], including 34 with chordoma, 14 with ASPS, 11 with SMARCA4-deficient malignant rhabdoid tumor (SMRT), 8 with desmoplastic small round cell tumor (DSRCT) and 31 with other histotypes. There were seven objective responses in ASPS (50%), three in SMRT (27%), one in DSCRT (12.5%), three in chordoma (8.8%) and one in other histotypes (3.2%). The greatest rates of PFS at 12 months were observed in ASPS (35.7%), chordoma (31.2%) and SMRT (18.2%).

SMARCA4-deficient thoracic sarcoma is a newly described entity of thoracic sarcomas that is associated with a poor prognosis. Partial responses to PD1/PDL1 blockade have been reported in PDL1-positive SMARCA4-deficient thoracic sarcomas with pembrolizumab [130,131,132], nivolumab [133] and atezolizumab plus bevacizumab combined with CT [134,135]. Marcrom et al. [136] reported a complete response of mediastinal clear cell sarcoma to pembrolizumab combined with RT. Yu et al. [137] reported a major response to pembrolizumab in an adult patient with an undifferentiated embryonal sarcoma of the liver. In metastatic myxofibrosarcoma, partial responses have been reported with pembrolizumab [138] and atezolizumab plus temozolomide [139]. These data, together with the data from AcSé trial, suggest that ICIs may be useful for certain subtypes of rare sarcoma, especially when selected by immune biomarkers such as the PDL1 expression.

A phase II trial with pembrolizumab in 17 patients with classic/endemic Kaposi sarcoma (KS) (71% of them pretreated with CT) showed 12 objective responses (ORR 70.1%), with 2 complete responses (CR) and 4 partial responses (PR) [89]. Interestingly, the lack of PDL1 expression on tumor and immune cells was associated with worse outcomes. Tabata et al. [140] reported a maintained response to ipilimumab plus nivolumab in a HIV-negative KS.

Nivolumab with and without ipilimumab was studied by D’Angelo et al. [90] in a phase II trial (Alliance A091401) in 85 patients with advanced sarcoma (BS and STS) after at least one previous line of treatment. The primary endpoint was the response rate. There were six objective responses in the combination group (16% vs. 5% in the monotherapy group), with a median duration of response of 6.2 months. There was a significant benefit for the combination group in terms of median PFS (4.1 m vs. 1.7 m) and median overall survival (OS) (14.3 m vs. 10.7 m). Responses occurred in UPS, LMS, myxofibrosarcoma, and angiosarcoma. Grade 3–4 toxicity was higher in the combined treatment (14 vs. 7%). Zhou et al. [141] reported two cases of PDL1-negative STS (DDLPS and myxofibrosarcoma) with long-term responses to ipilimumab plus nivolumab.

In the first-line setting, a retrospective study of nivolumab with or without ipilimumab in PDL1-positive STS found a significant benefit for the combination group in terms of ORR (13% vs. 7%), median PFS (4.1 m vs. 2.2 m) and median OS (12.2 m vs. 9.2 m) [142]. A similar ORR (15%) was observed in a retrospective study with ipilimumab plus nivolumab for advanced STS [143]. The combination ipilimumab plus nivolumab is also being evaluated by a phase II trial in patients with pre-treated classic KS (NCT03219671), with a promising ORR of 50% in an interim analysis [144].

A recent phase II trial has analyzed the combination of anti-CTLA4 tremelimumab plus anti-PDL1 durvalumab in 57 patients with advanced STS and BS after at least one line of systemic treatment [91]. The most represented subtypes were ASPS (18%), LPS (10%) and vascular tumors (18%). The median PFS and OS for all subtypes were 2.8 months and 21.6 months, respectively. The median PFS at 12 weeks (the primary endpoint) was 49% (95% CI 36–61). Global ORR was 12%, though there were significant differences among histological subtypes, with the greatest benefit observed in the ASPS subgroup (ORR 40%, including 2 CR). The authors concluded that tremelimumab and durvalumab is an active treatment for advanced sarcoma.

Lewin et al. have reported two ASPS with a durable response to durvalumab alone or combined with tremelimumab, confirming their activity in this subtype [145]. A randomized phase II trial (MEDI-SARC) is currently comparing durvalumab plus tremelimumab to CT (doxorubicin) in naïve-treatment STS [146]. Anti-PDL1 avelumab has been tried in recurrent osteosarcoma with negative results [147].

The combined blockade of PD1 and LAG3 is another encouraging strategy. A basket phase II study of anti-PD1 spartalizumab plus anti-LAG3 LAG525 in advanced solid tumors included a cohort of 10 patients with advanced sarcoma, reporting a disease control rate (DCR) of 40% at 24 weeks [148]. The combination nivolumab plus anti-LAG3 relatlimab, clearly superior to nivolumab alone in untreated melanoma according to the phase III trial RELATIVITY-047 [149], is being evaluated in metastatic STS by a phase II study (NCT04095208).

According to these results, there is not enough evidence to support the use of ICIs as monotherapy in the first-line setting of advanced sarcoma since the previous trials included mainly pre-treated patients. However, given the scarcity of therapeutic alternatives for CT-refractory patients, the authors consider that treatment with ICIs might be considered after progression to standard CT, especially in patients with immunogenic subtypes of STS (classic/endemic KS, UPS, synovial sarcoma and dedifferentiated liposarcoma). Dual blockade of CTLA4 and PD(L)1 with nivolumab plus ipilimumab or durvalumab plus tremelimumab seems to offer higher response rates and may be preferable to single-agent immunotherapy in fit patients. Although imperfect, predictive biomarkers such as high TMB, high PDL1 expression or dense lymphocytic infiltration may help clinicians decide to use ICIs in this subgroup of patients.

Several additional trials with monotherapy or a combination of ICIs are currently ongoing [25], such as anti-PDL1 atezolizumab alone or plus bevacizumab in ASPS (NCT03141684) and ipilimumab plus nivolumab in classical Kaposi sarcoma (NCT03219671).

### 3.2. Combination of ICIs and Conventional Chemotherapy

The combination of immunotherapy with CT is a promising approach to enhance antitumor activity in sarcoma. The DNA damage caused by cytotoxic drugs results in cell death, with the subsequent release of DAMPs and proteins that work as ‘danger signals’, upregulating PD1 and enhancing the activity of effector lymphocytes [150].

A phase II trial that studied pembrolizumab in combination with doxorubicin in 30 patients with unresectable STS, with no previous anthracycline therapy, showed interesting results, with a DCR of 80% and a global ORR of 36.7% [92]. The subtypes with the highest ORR were UPS (4/4 patients), epithelioid angiosarcoma (1/1 patient), leiomyosarcoma (4/10 patients) and liposarcoma (2/7 patients). The median PFS and OS were 5.7 months and 17 months, respectively. In this study, PDL1 expression was associated with improved ORR. The authors concluded that the combination of pembrolizumab and doxorubicin has manageable toxicity and promising activity in advanced STS.

Pollack et al. [93] performed another phase I/II trial with pembrolizumab plus doxorubicin in 37 anthracycline-naïve patients with advanced STS, reporting an ORR of 19%, a median PFS of 8.1 months and a median OS of 27.6 months. Similarly, durable partial responses were observed in two of three patients with UPS and two of four patients with DDLPS. A retrospective study of pembrolizumab plus doxorubicin in 21 patients with STS showed a similar DCR (71.4%) [151]. As in the clinical trials, patients with UPS, synovial sarcoma and angiosarcoma showed higher response rates.

The combination of pembrolizumab plus metronomic cyclophosphamide showed limited activity in the phase II trial PEMBROSARC [79], with just 1 partial response among 50 advanced STS. Interestingly, the response was observed in the only case, a solitary fibrous tumor, with a PDL1 expression >10% in immune cells.

Italiano et al. [152] demonstrated that the presence of tertiary lymphoid structures (TLS) was a powerful predictor of response among the patients from the PEMBROSARC trial, with an ORR of 40% in TLS+ and 26.7% in TLS- tumors. New strategies are needed to induce TLS neogenesis and sensitize TLS-negative tumors to immunotherapy. In fact, a cohort of 20 TLS- patients from the PEMBROSARC study were treated with pembrolizumab, low-dose cyclophosphamide and intra-tumoral injection of the toll-like receptor 4 (TLR4) agonist G100, which would potentially enhance the immune response against a TLS-negative TME [153]. G100 seemed to modulate the TME, increasing TILS infiltration, though the ratio CD8+/FoxP3+ decreased in 11 out of 14 assessable cases, suggesting a predominant recruitment of T-regs-, a finding that may explain the modest clinical results (PFS of 11.8% at 6 months).

A phase I/II clinical trial (SAINT) has analyzed the dual CTLA4/PD1 blockade with ipilimumab plus nivolumab added to trabectedin in advanced STS, with encouraging results. Among 79 patients enrolled in the phase II study, there was a DCR of 87.3% (6 CR, 14 PR, 49 SD), with a median PFS of 6.7 months and a median OS of 24.6 months [94]. A retrospective study of nivolumab alone combined with trabectedin in 28 pre-treated STS found a DCR of 72.7% (4 PR and 12 SD among 22 assessable patients), with a median PFS at 6 months of 68.2% [154].

The results of trabectedin combined with dual immune blockade (ipi/nivo) seem better than its combination with nivolumab alone, which obtained modest results in the phase II trial NITRASARC (2 PR among 25 evaluable patients, with an mPFS of 4 months) [95].

The phase I trial GEMMK [96] is studying the combination of pembrolizumab plus gemcitabine in STS; among 13 patients included, there were 11 leiomyosarcomas (LMS) (with 8 SD and 3 PD) and 2 UPS (with 2 PR). A phase II trial of pembrolizumab plus eribulin in STS (NCT 03899805) has reported preliminary data from the LMS cohort (19 patients), with limited efficacy (ORR 5.3% and DCR 26.3% after 12 weeks) [97].

A phase I/II study has evaluated anti-PDL1 avelumab combined with trabectedin in 33 patients with advanced liposarcoma and LMS, with an ORR of 13%, a DCR of 56% (3 PR, 10 SD) and a median PFS of 8.3 months [98]. Trabectedin has also been combined with durvalumab by a phase Ib study (TRAMUNE) in 16 patients with STS, with an ORR of 7% and a 6-month PFS of 28.6% [99].

The heterogeneity in the selection of patients may explain the significant differences in the results of these studies with combination strategies. The best results seem to be obtained with the early use of ICI+CT, particularly with dual immune blockade (CTLA4/PD1). As monotherapy with ICIs, the combination of ICI+CT seems to obtain better response rates in certain histologic subtypes of STS (such as liposarcoma and UPS). In our opinion, there are promising data supporting the combination of nivo/ipi plus trabectedin for advanced STS in anthracycline-refractory patients, as well as pembro plus doxorubicin in anthracycline-naïve patients. Phase III trials comparing these schemes to the standard treatment would be helpful to confirm these results and incorporate the ICI+CT combinations into the first-line of systemic treatment.

Several ongoing phase I/II clinical trials are studying combinations of different cytotoxic drugs with pembrolizumab (NCT 03899805, NCT 03123276, NCT 04332874), nivolumab (NCT 04535713, NCT 03590210) and durvalumab (NCT 03802071) [25]. Their results will hopefully help us identify proper ICI/CT synergistic combinations with manageable toxicity.

### 3.3. Combination of ICIs and Tyrosine–Kinase Inhibitors

Tyrosine–kinase inhibitors (TKIs) are progressively gaining ground in CT-refractory sarcoma. In addition to blocking the immune-suppressive effect of VEGF [83], multi-target TKIs, especially lenvatinib and cabozantinib, seem to decrease the arrival of MDSCs/TAMs to the TME, and increase the infiltration of dendritic cells, NK cells and CD8+ lymphocytes [155]. This favorable immune-modulating effect provides a rationale for their combination with ICIs.

The combination of nivolumab and sunitinib in advanced sarcoma has been studied by the phase I/II trial IMMUNOSARC, including 40 patients with BS and 50 patients with STS. In the BS cohort, there was 1 CR (2.5%), 1 PR (2.5%) and 22 SD (55%) among 40 assessable patients, with a median PFS of 3.7 months [100]. In the STS cohort, there was 1 CR (2.3%), 3 PR (7%) and 26 SD (60%) among 43 evaluable patients, with a median PFS of 5.9 months [101].

A phase II clinical trial [102] has evaluated pembrolizumab plus axitinib in 33 patients with advanced sarcoma, mainly STS (12 ASPS, 6 LMS, 5 UPS, 2 DDLPS and 8 other subtypes, including 2 BS). 51% of patients had received prior treatment with TKIs, and 15% of them with immunotherapy. Among 32 assessable patients, the ORR was 25% (8 PR) and the median PFS was 4.7 months in the intention-to-treat analysis. Six of the eight patients with a PR had ASPS, a subgroup in which this combination seems especially active (ORR 50%), as confirmed by a post-hoc analysis, median PFS of 12.4 months in the ASPS subgroup. Dorman et al. [156] reported a case of ASPS with a durable (15 months) response to pembrolizumab plus axitinib, in consonance with the previous data. The rarity of ASPS, which comprise just 1% of STS, and its unique biology, which seems to make it especially responsive to ICI/TKI combinations, have led some authors to warn about the potential skew of unselected STS studies that include a high proportion of this subtype.

The phase II trial APFAO [103] evaluated anti-PD1 camrelizumab in combination with apatinib in 43 CT-refractory osteosarcomas, with an ORR of 20.9% and a median PFS of 6.2 months. A PDL1 tumor proportion score >5% and lung metastasis correlated with longer PFS. A retrospective study in 33 STS (8 ASPS, 5 LMS, 3 UPS and 17 others) showed an interesting activity of this combination added to local therapy with radiofrequency ablation or transarterial chemoembolization [157].

Durvalumab plus pazopanib have also shown encouraging activity in advanced STS in a phase II trial with 47 patients, with an ORR of 28.3% (1 CR and 12 PR), a median duration of response of 11 months and a median PFS of 8.6 months [104]. Combinations of pazopanib with PD1 inhibitors have been explored by retrospective studies. Paoluzzi et al. [158] treated 28 patients (24 STS and 4 BS) with nivolumab (plus pazopanib in 18 cases), with a DCR of 50% and 3 PR (one CS, one epithelioid sarcoma and one maxillary osteosarcoma, last two patients on pazopanib). Arora et al. [159] reported a sustained response to pembrolizumab plus pazopanib in a patient with advanced UPS.

The combination of avelumab plus regorafenib has been tried by a phase II basket trial in advanced solid tumors (REGOMUNE), with an STS cohort including 49 patients (22 LMS, 9 synovial sarcomas, 4 liposarcomas, 4 UPS and 10 other subtypes) [105]. The results were similar to other combinations of ICI + TKIs, with 4 PR among 43 evaluable patients and a DCR of 48.8%.

Given the known immune-suppressive role of IDO1 in sarcoma, explained above in detail, there is a rationale for combining ICIs with TKI against IDO1. A phase II trial with pembrolizumab plus IDO1 inhibitor epacadostat showed modest efficacy in 29 pre-treated advanced sarcomas (ORR 3%, DCR 48%) [106]. A phase I trial has shown promising results of pembrolizumab plus olaratumab PDGFR inhibitor in 28 patients with advanced STS (ORR 21.4%, DCR 53.5%) [107].

In our opinion, the combination of ICI/TKI (nivolumab/sunitinib, pembrolizumab/axitinib, durvalumab/pazopanib or avelumab/regorafenib) should be considered in patients with advanced STS after progression to standard CT. The authors believe that further research is required to evaluate the benefits of these combinations compared to ICI or TKI monotherapy. It would also be interesting to evaluate this strategy in the first-line setting of patients *unfit* for anthracycline-based CT.

Several ongoing trials will assess the combination of ICIs with TKIs and other antiangiogenic drugs [25], including lenvatinib plus pembrolizumab (NCT 04784247), cabozantinib plus nivolumab and ipilimumab (NCT 04551430), anlotinib plus toripalimab (NCT 04172805), olaparib or cediranib plus durvalumab (NCT 03851614), epacadostat plus pembrolizumab (NCT 03414229), bevacizumab plus atezolizumab (NCT 03141684), tivozanib plus atezolizumab (NCT 05000294), dasatinib plus ipilimumab (NCT 01643278) and axitinib plus avelumab (NCT04258956).

### 3.4. Combination of ICIs and Other Agents

Various studies have explored the combination of ICIs with other immunomodulatory therapies. One of the most promising agents is talimogene laherparepvec (T-VEC), a modified immune-enhanced herpes simplex virus (HSV) type 1, engineered for intratumoral injection. It contains the coding sequence for granulocyte–macrophage colony-stimulating factor (GM-CSF) and proinflammatory cytokines, inducing an immune response against tumor cells. Preoperative RT plus T-VEC has been studied in 30 locally advanced STS, with modest results [160].

A phase II trial has evaluated T-VEC plus pembrolizumab in 20 cases of advanced sarcoma (after at least one standard therapy), with promising results (7 PR, with a median duration of response of 56 weeks, and 7 SD) and an acceptable safety profile. The most represented subtypes were LMS (25%), cutaneous angiosarcoma (15%) and UPS (10%) [108]. The mean TIL score was higher in responsive patients. A phase II trial has analyzed T-VEC plus nivolumab and trabectedin in 36 advanced sarcomas, including chordoma and desmoid tumor heavily pre-treated patients, with a median of 4 previous lines, with encouraging results: 3 PR, 27 SD (DCR 86.1%) and a median PFS of 5.5 months (being the median PFS of the immediately prior therapy 2.0 months) [109].

Bempegaldesleukin (NKTR-214), an interleukin-2 (IL-2) agonist, increases TIL infiltration and has shown encouraging activity in several refractory tumors [161], setting a rationale for its addition to PD1 inhibitors. A pilot study combining this agent with nivolumab in 84 patients showed positive results in angiosarcoma (3 PR out of 8), ASPS (1 PR out of 4) and UPS (2 PR out of 10) [110]. There was a correlation between the ORR, CD8+ TIL infiltration and PDL1 expression. Interestingly, the reduced expression of the Hedgehog signaling pathway was associated with better clinical outcomes, suggesting that the Hedgehog pathway enhances immune-suppressive mechanisms as it has been found in breast cancer [162] and skin basal cell carcinoma [163] and setting a rationale for combining immunotherapy with Hedgehog inhibitors.

Oleclumab is a monoclonal antibody that selectively binds and blocks the activity of CD73, a key enzyme for the extracellular generation of adenosine, which is a relevant biochemical component of the immunosuppressive TME [164]. An ongoing phase II study is evaluating the combination of durvalumab and oleclumab in specific sarcoma subtypes selected according to CD73 staining on the TME cells [165]. A basket phase II trial (CAIRE) is evaluating durvalumab plus tazemetostat, a molecule that inhibits EZH2, which leads to a functional alteration of Tregs and favors an effector-like profile in advanced solid tumors, including a cohort for STS [166].

Another approach is the modulation of immune cells in vivo. LV305 is a modified lentivirus-based vector designed to selectively transduce dendritic cells and promote the expression of NY-ESO1, unleashing an immune response against NY-ESO1-expressing cells. Somaiah et al. [111] tried LV305 in 24 patients (13 SS, 6 MRCL and 5 other subtypes), reporting 1 PR (SS) and 14 SD. NY-ESO1 expression was >75% in 67% of the subjects. A phase I study has evaluated avelumab plus SNK01, a therapy based on autologous modified NK cells, with enhanced cytotoxicity, in 15 patients with advanced sarcoma (6 LMS, 2 osteosarcoma and 7 other subtypes) [113], with 2 PR and 3 SD.

Other strategies to modulate the immune-suppressive TME of sarcoma have been explored in preclinical studies and early trials. Colony-stimulating factor 1 receptor (CSF1R) signaling regulates the infiltration of MDSCs and promotes their differentiation towards an M2 phenotype [167]. A phase Ib study has explored the combination of durvalumab with DCC-3014 (vimseltinib), a selective inhibitor of CSF1R, in 13 patients with advanced STS (7 leiomyosarcoma, 2 UPS, 2 DDLPS, 1 synovial sarcoma and 1 liposarcoma) [112]. There was a median decrease of 26.9% in circulating MDSCs, with disease stabilization in three patients.

### 3.5. Therapeutic Vaccines

Cancer vaccines are usually based on the exogenous administration of selected tumor antigens combined with adjuvants to induce the activity of APCs, mainly dendritic cells, aiming to stimulate the adaptive immune system against cancer cells [168].

Sato et al. [169] demonstrated that peptides derived from *SYT-SSX* fusion genes, resultant from chromosomal translocation t(X:18) specific to synovial sarcoma (SS), are recognized by circulating CD8+ T-cells in HLA-A24+ patients, and induce tumor-specific cytotoxic responses. They developed an *SYT-SSX* peptide vaccine and tried it on six patients with advanced SS, with no response [114]. However, they performed a second study adding interferon-alfa to the vaccine in 12 patients, observing 6 SD and a transient response in 1 patient [170].

Takahashi et al. [115] conducted a phase II clinical trial with a personalized vaccine in 20 patients with refractory BS and STS. A maximum of four HLA-matched peptides were selected, based on their high peptide-specific IgG responses in pre-vaccination plasma, and administered to each patient weekly for 6 weeks and each 2 weeks thereafter; six patients achieved SD, including one minor response in a malignant fibrous histiocytoma, and one durable SD (33 months) in a mediastinal SS. The authors concluded that this treatment could be feasible for the vast majority of refractory sarcoma patients, with high rates of immunological responses.

Neoantigens such as cancer–testis antigens (CTA), aberrantly expressed in a high percentage of high-grade sarcoma, mainly MAGE and NY-ESO, as explained above in detail, might also be used to design personalized vaccines [32]. A phase I/II trial has evaluated an autologous dendritic-cell vaccine based on CTA (CaTeVac) in 74 patients with advanced STS, with a cohort receiving the vaccine as adjuvant or maintenance treatment after the first or second line of systemic therapy and another cohort receiving it in monotherapy after at least one CT line [116]. Median OS was 24.4 months in the first cohort and 14.2 months in the second one, suggesting a positive impact on overall survival. A phase I study tried a dendritic cell vaccine-targeting MAGE-A1, MAGE-A3 and NY-ESO1 combined with decitabine in 10 children with neuroblastoma, Ewing’s sarcoma, osteosarcoma and rhabdomyosarcoma. Six of nine patients developed a response to MAGE-A1, MAGE-A3 or NY-ESO-1 peptides post-vaccine, concluding that the chemoimmunotherapy approach using DAC/DC-CT vaccine is feasible [171].

More recently, Chawla et al. [117] have published a phase II randomized study of CMB305 and atezolizumab compared with atezolizumab alone in STS expressing NY-ESO1. CMB305 is a vaccination regimen designed to prime the CD8+ T-cell population specific for NY-ESO1 and then use a TLR4 agonist to unleash the antitumor immune response. The study recruited 89 patients with SS and myxoid liposarcoma (MLS). Though the PFS increase was not significant, the patients treated with the combination acquired a higher rate of NY-ESO1-specific T-cells, which was associated with longer OS in a post hoc analysis.

In addition to vaccines based on specific peptides, a potential approach to induce tumor recognition is the production of vaccines derived from whole tumor cells combined with immune-enhancing adjuvants (such as IFN-γ and GM-CSF), with interesting preliminary data in a cohort of STS [172].

### 3.6. Adoptive Cell Therapy

Adoptive cell therapy (ACT) encompasses several strategies to improve the activity of autologous T cells, which are obtained through leukapheresis and genetically engineered to overcome tumor immune evasion. Modified T cells are reinfused into the patient, usually in combination with adjuvant IL-2, after lymphodepleting CT (commonly fludarabine plus cyclophosphamide). ACT includes chimeric antigen receptor (CAR) T cell therapy, engineered TCRs and TIL therapy [173].

#### 3.6.1. Engineered TCR

In sarcoma, the aberrant expression of CTA not present in normal cells makes TCR therapy particularly attractive since T cells can be equipped with newly engineered TCRs transduced through retroviral vectors that enable them to target these specific neoantigens.

Robbins et al. [118] conducted a phase II trial with letetresgene autoleucel (Lete-cel), a therapy based on T cells transduced with a TCR against NY-ESO1, plus adjuvant IL-2- in HLA-A02 patients with metastatic melanoma and synovial sarcoma. Among 18 patients with NY-ESO1+ SS, 11 objective responses were documented (61%), including 1 CR and 10 PR (ranging from 3 to 47 months). The estimated overall 3-year OS rate was 38%. The authors concluded that the adoptive transfer of autologous T cells transduced with a retrovirus encoding a TCR against an HLA-A*0201 restricted NY-ESO-1 epitope can be an effective therapy for synovial cell sarcomas. Considering that all the patients were heavily pre-treated and refractory to standard CT, the authors consider it improbable that the lymphodepleting therapy could significantly contribute to the clinical responses.

More recently, Lete-cel has been evaluated in 45 patients with advanced SS after standard first-line CT, enrolled in 4 cohorts according to NY-ESO1 tumor expression [119]. Objective responses were documented in all cohorts, with a total of 1 CR and 14 PR (ORR 20–50%). The median PFS ranged from 8.6 to 22.4 weeks between cohorts. A post-hoc analysis revealed that responders had higher pre-infusion levels of IL-15, which may be used as a predictive biomarker [174]. NY-ESO1-targeted TCR therapy is also being evaluated in myxoid/round cell liposarcoma (MRCL) [175].

A first-in-human study with T cells equipped with afamitresgene autoleucel (Afami-cel), a genetically engineered autologous specific peptide enhanced affinity receptor (SPEAR) targeting MAGE-A4, reported 4 PR and 3 SD among eight assessed patients with advanced SS [120]. Following these results, Afami-cel was evaluated by a phase II trial (SPEARHEAD-1) in 32 eligible patients (HLA-A02) with MAGE-A4-expressing STS (87.5% SS, 12,5% MLS) [121]. Among 25 evaluable subjects (23 SS and 2 MLS), there were 2 CR, 8 PR and 11 SD (DCR 84%). The safety profile was favorable, with mainly low-grade cytokine release syndrome (CRS) and reversible hematologic toxicities due to lymphodepleting CT. A pooled efficacy analysis of phase I and II trials with Afami-cel revealed that the baseline tumor burden, prior systemic treatment history and MAGE-04 expression levels are potential predictors of response [176].

#### 3.6.2. CAR T Cells

The use of CAR T cells has also been explored in sarcoma, with the advantage over TCR therapy of not being restricted to HLA-A02 carriers. In contrast to hematologic malignancies, the efficacy of CAR T cells in solid tumors is limited by their intense antigenic heterogeneity derived from their polyclonal expansion and accumulative mutations, which makes it hard to find homogeneously expressed targets, particularly without unacceptable off-tumor toxicity [177].

Human epidermal growth factor receptor 2 (Her2) is the most studied target for CAR T cells in sarcoma. A phase I/II trial evaluated Her2-CAR T cells in 19 Her2+ sarcoma (16 of them osteosarcoma), with modest activity (4 SD ranging from 12 weeks to 14 months). Her2-CAR T cells persisted for at least 6 weeks in seven of the nine evaluable patients. Three of these patients had their tumors removed, with one showing ≥90% necrosis. The median overall survival was 10.3 months (ranging 5.1 to 29.1 months) [122].

In a phase I trial with Her2-CAR T cells in 10 Her2+ sarcoma, there were 2 long-term CR, 1 in osteosarcoma and 1 in rhabdomyosarcoma, added to 3 SD [123]. A phase I study is currently using the combination of Her2-CAR T therapy with PD1 blockade pembrolizumab or nivolumab in advanced Her2+ sarcoma (NCT 049955003).

Other phase I trials are currently evaluating CAR T cells targeting epidermal growth factor receptor (EGFR) (NCT 03618381) and tumor antigen GD2 (NCT 04539366, NCT 03635632, NCT 02107963, NCT 03721068) [177]. Preclinical studies have also suggested a promising activity of CAR T cells targeting the type I insulin-like growth factor receptor (IGF1R) and the tyrosine–kinase-like orphan receptor 1 (ROR1), which are highly expressed in sarcoma cell lines [178]. Other potential targets for CAR T therapy in sarcoma are CD44v6 [179] and NK cell activating receptor group 2-member D ligand (NKG2DL) [180,181].

#### 3.6.3. TIL Therapy

TIL therapy is based on the extraction of tumor-infiltrating lymphocytes for ex vivo expansion and reinfusion to the patient, after lymphodepleting conditioning, in combination with immune-enhancing adjuvants such as IL-2. It has an exciting potential, being the only ACT-using cells with multiple TCR clones able to cover the antigenic heterogeneity of solid tumors in contrast to engineered TCRs and CAR T cells, which target specific antigens [182]. Up to date, TIL therapy has been mainly developed in melanoma with a positive phase III trial versus ipilimumab in the first-line setting [183] and has promising results in cervical cancer [184,185,186], non-small cell lung cancer [187], ovarian cancer [188], colorectal cancer [189], breast cancer [190] and cholangiocarcinoma [191].

Mullinax et al. [192] demonstrated the feasibility of TIL therapy in STS. They successfully propagated TILs from 70 STS surgical specimens, cocultured them with tumor cells, and expanded them using a rapid expansion protocol, observing that nearly all specimens generated TILs, mainly CD3+, which were responsive to the autologous tumor. Ko et al. [193] published another preclinical study that suggests that certain sarcoma subtypes can potentially yield an appropriate number of cells for TIL therapy.

Zhou et al. [194] conducted a retrospective study of 60 patients with CT-resistant metastatic osteosarcoma who were treated with TIL therapy plus nivolumab. Of the patients, 83.3% had lung metastasis, and 83.3% had presented a poor response to neoadjuvant CT. The results were encouraging, with an ORR of 36.7% (2 CR and 20 PR), a DCR of 80% and a median PFS of 5.8 months. An infusion of ≥ 5 × 10 [9] TIL cells, a percentage of CD8+ TIL ≥ 60% and a percentage of CD4+/FoxP3+ TIL <20% were significantly associated with response to treatment. Overall, OS was 23.7 months in responders versus 8.7 months in non-responders (*p* < 0.0001).

Despite these interesting preclinical and retrospective data, the use of TIL therapy in sarcoma is not endorsed by any clinical trial to date. A phase I study of TILs in STS (NCT 04052334) and two phase II basket trials, one with a cohort for carcinosarcoma (NCT 03610490) and another with a cohort for STS (NCT 03935893), are ongoing [177]. Further research is required to understand how to overcome the challenges still faced by TIL therapy, such as the negative impact of the immunosuppressive TME or its application in ‘cold’ tumors, including many sarcoma subtypes.

## 4. Conclusions

Immunotherapy is progressively acquiring a role in the treatment of advanced sarcoma, though the biological heterogeneity among histologic subtypes impacts their clinical response. Globally, soft tissue sarcomas (STS) are more responsive than bone sarcomas (BS). Undifferentiated pleomorphic (UPS), alveolar soft part (ASPS), synovial and Kaposi sarcomas seem especially immunogenic, followed by some liposarcoma and leiomyosarcoma. The presence of tertiary lymphoid structures (TLS), a high density of infiltrating CD8+ T cells and a high PDL1 expression in cells of the TME have been identified as prognostic and predictive biomarkers.

Immune checkpoint inhibitors (ICI) have poor results in monotherapy, except for certain subtypes. The SARC028 trial is the most relevant study with an ICI alone (pembrolizumab), with an ORR of 40% in UPS and 20% in liposarcoma and chondrosarcoma but <10% in the other subtypes. Kaposi and other rare forms of sarcoma, such as chordoma and SMARCA4-deficient rhabdoid tumors, might also benefit from PD1 inhibitors. Dual blockade with ipilimumab plus nivolumab showed superiority compared to single immunotherapy (Alliance A091401), and durvalumab plus tremelimumab seems especially active in ASPS.

Multiple studies have evaluated ICI combined with conventional CT, trying to achieve a synergistic effect. Pembrolizumab plus doxorubicin has positive results in anthracycline-naïve STS, and some other combinations, ipi/nivo plus trabectedin, pembro plus gemcitabine and avelumab plus trabectedin, have promising results in CT-refractory STS. The IMMUNOSARC trial is the main study combining ICI with tyrosine–kinase inhibitors (nivolumab plus sunitinib), reaching a DCR >60% both in STS and BS, though other combinations have shown similar outcomes in phase I and II trials. ICI has also been combined with immunomodulatory agents, with promising results of oncolytic viral therapy (T-VEC) added to PD1 blockade.

Going beyond checkpoint inhibitors, therapeutic vaccines and adoptive cell therapy are incorporated into the arsenal of immunotherapy, with encouraging results. Engineered TCR targeting aberrant neoantigens NY-ESO1 (Lete-cel) and MAGE-A4 (Afami-cel) have shown ORR 40–60% in synovial sarcoma, with a favorable safety profile. Her2-targeted CAR-T cells and TIL therapy have promising preclinical and retrospective data in advanced sarcoma, with the potential benefit of their combination with PD1 inhibitors. Further research is still needed to overcome the theoretical and practical obstacles faced by adoptive cell therapy, especially in such aggressive tumors with a powerful immunosuppressive microenvironment.

## Figures and Tables

**Figure 1 cancers-15-02287-f001:**
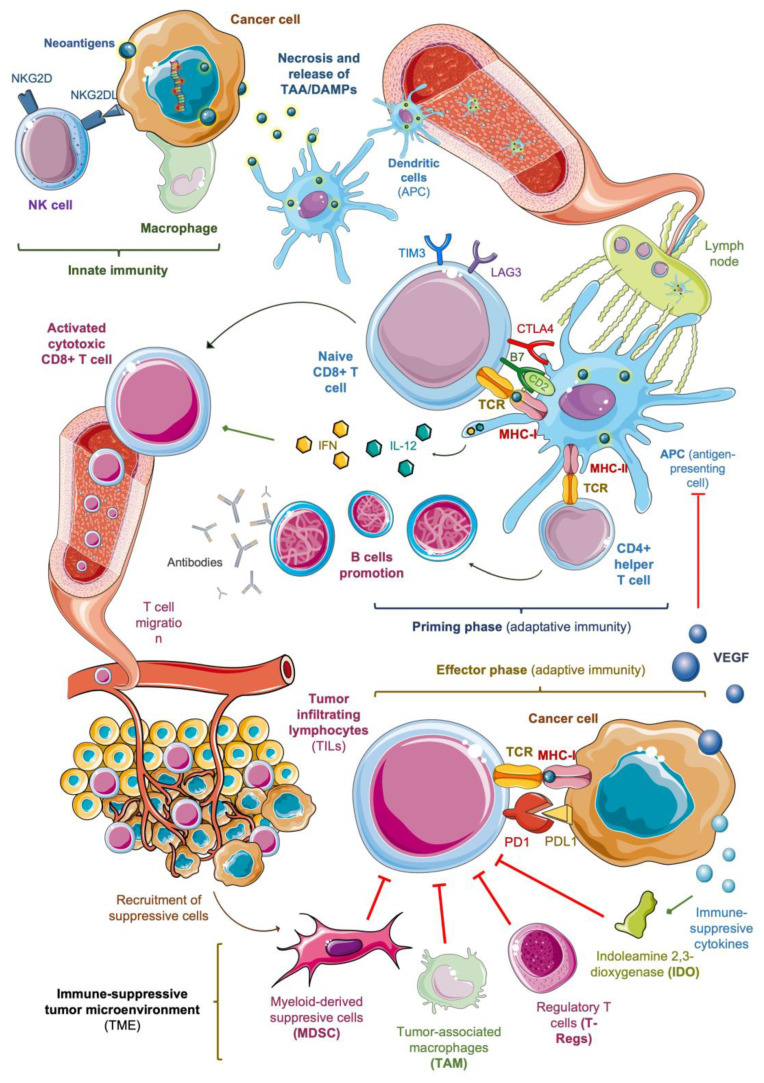
Mechanisms of anti-tumor response and potential immune biomarkers in solid tumors. NK: natural killer; TAA: tumor-associated antigen; DAMPs: damage-associated molecular patterns; TCR: T cell receptor; MHC: major histocompatibility complex; IL-12: interleukin-12; IFN: interferons; VEGF: vascular endothelial growth factor.

**Table 1 cancers-15-02287-t001:** Published results of immunotherapy in sarcoma. AEs: adverse effects; irAEs: immune-related adverse effects; ORR: objective response rate (RECIST criteria); mPFS: median progression-free survival; mOS: median overall survival; m: months; w: weeks; PR: partial response; SD: stable disease; DCR: disease control rate (PR+SD); mDR: median duration of response; NA: not available; AST: aspartate aminotransferase; ALT: alanine aminotransferase; BS: bone sarcoma; STS: soft-tissue sarcoma; OST: osteosarcoma; ES: Ewing sarcoma; CS: chondrosarcoma; LMS: leiomyosarcoma; LPS: liposarcoma; SS: synovial sarcoma; UPS: undifferentiated pleomorphic sarcoma; KS: Kaposi sarcoma; ALP: alkaline phosphatase; CK: creatine kinase; mCP: metronomic cyclophosphamide; AS: angiosarcoma; ASPS: alveolar soft-part sarcoma; DDLPS: dedifferentiated liposarcoma; PPS: palmar-plantar syndrome; LDH: lactate dehydrogenase; MDSCs: myeloid-derived suppressor cells; MRCL: myxoid/round cell liposarcoma; NY-ESO1: New York esophageal squamous cell carcinoma 1; MAGE-A4: melanoma-associated antigen A4; DC: dendritic cells; MLS: myxoid liposarcoma; CRS: cytokine release syndrome; RMS: rhabdomyosarcoma; adj: adjuvant; maint: maintenance; mono: monotherapy.

Clinical Trial	Agent	Tumor	N	Age Range	Outcomes	Reported G3/G4 AEs
Immune checkpoint inhibitors (ICI)						
Maki et al. (phase I/II) [85]	Ipilimumab	Synovial sarcoma	6	23–57	ORR 0% mPFS 1.5 m mOS 8.8 m	Nausea (50%), diarrhea (33.3%), lymphopenia (33.3%), hyperbilirubinemia (16.7%), thrombopenia (16.7%)
Merchant et al. (phase I) [86]	Ipilimumab	Pediatric sarcoma	17	2–17	ORR 0%; DCR 17.6% (3 SD) mPFS/mOS: NA	Diarrhea (9%), AST/ALT increase (6%), endocrinopathies (3%), other irAEs (9%)
Ben-Ami et al. (phase II) [87]	Nivolumab	Uterine leiomyo-sarcoma	12	29–73	ORR 0% mPFS 1.8 m; mOS NA	Lipase/amylase increase (8.3%), fatigue (8.3%), abdominal pain (8.3%)
Tawbi et al. (phase II) (SARC028) [54]	Pembrolizumab	BS cohort (22 OST, 13 ES, 5 CS)	40	16–70	mPFS 8 w; mOS 52 w OST: ORR 5%; DCR 32% (1 PR, 6 SD) ES: ORR 0%; DCR 15% (2 SD) CS: ORR 20%; DCR 40% (1 PR, 1 SD)	Interstitial nephritis (2%), infectious pneumonia (2%), bone pain (2%), pleural effusion (2%), hypoxia (2%)
		STS cohort (10 LMS, 10 LPS, 10 SS, 10 UPS)	40	18–81	mPFS 18 w; mOS 49 w LMS: ORR 0%; DCR 60% (6 SD) / LPS: ORR 20%; DCR 60% (2 PR, 4 SD)/SS: ORR 10%; DCR 30% (1 PR, 2 SD)/UPS: ORR 40%; DCR 70% (1 CR, 3 PR, 3 SD)	Pulmonary embolism (2%), adrenal insufficiency (2%), pneumonitis (2%)
Blay et al. (phase II) (AcSé) [88]	Pembro-lizumab	Advanced rare sarcoma	98	>18 (NA)	ORR 15.3%; DCR 49% (1 CR, 14 PR, 33 SD); mDR 8.2 m; mPFS 2.8 m; mOS 19.7 m	NA
Delyon et al. (phase II) [89]	Pembrolizumab	Classic/ endemic KS	17	NA	ORR 70.1%; DCR 88% (1 CR, 10 PR, 4 SD)	Reversible acute cardiac decompensation (6%)
D’Angelo et al. (phase II) (Alliance A091401) [90]	Nivo/ipi vs. ipi	Advanced sarcoma (BS and STS)	85	21–81	ORR 16% vs. 5%; mDR 6.2 mmPFS 4.1 m vs. 1.7 m mOS 14.3 m vs. 10.7 m	Pain (7% vs. 5%), thrombopenia (0% vs. 2%), pulmonary edema (2% vs. 0%), respiratory failure (5% vs. 5%), skin infection (2% vs. 0%), intestinal obstruction (2% vs. 2%), spinal fracture (0% vs. 2%), thrombo-embolic event (2% vs. 2%), urinary tract infection (7% vs. 2%), urinary obstruction (0% vs. 5%), fistula (2% vs. 0%), vomiting (0% vs. 2%)
Somaiah et al. (phase II) [91]	Durva-lumab + tremeli-mumab	Advanced sarcoma (BS and STS)	57	35–59	mPFS 2.8 m; mOS: 21.6 m; PFS at 12 m (all): 28%; PFS at 12 m (ASPS): 80% ORR (irRECIST) (all): 12%; ORR (irRECIST) (ASPS): 40%	Lipase increase (7%), pneumonitis (6%), colitis (6%), myocarditis (4%), autoimmune disorders (4%), endocrine disorders (2%), diarrhea (2%), gastrointestinal disorders (2%), lung infection (2%), ALP increase (2%), amylase increase (2%), myositis (2%)
Immune checkpoint inhibitors (ICI) + conventional CT						
Livingston et al. (phase II) [92]	Pembro + doxorubicin	Anthracy-cline naïve STS	30	NA	ORR 36.7%; DCR 80% (1 CR, 10 PR, 13 SD) mPFS 5.7 m; mOS 17 m PFS at 6 m: 44% PFS at 12 m: 62%	Neutropenia (36.7%), anemia (26.7%), febrile neutropenia (16.7%), arthralgia (13.3%), lymphopenia (13.3%), nausea (13.3%), fatigue (10.0%), hyponatremia (10.0%), vomiting (10.0%), lung infection (10.0%), muscle weakness (10.0%)
Pollack et al. (phase I/II) [93]	Pembro + doxorubicin	Anthracy-cline naïve STS	37	25–80	ORR 19%; DCR 78% (7 PR, 22 SD); mPFS 8.1 m; mOS 27.6 m PFS at 12 m: 27%	Neutropenia (27.0%), oral mucositis (8.1%), anemia (5.4%), febrile neutropenia (5.4%), lymphopenia (5.4%), ejection fraction decrease (5.4%), anorexia (5.4%), diarrhea (2.7%), hypothyroidism (2.7%), nausea (2.7%), weight loss (2.7%)
Toulmonde et al. (phase II) [79]	Pembro + mCP	Advanced STS	50	18–84	ORR 2%; DCR 34% (1 PR, 16 SD); PFS at 6 m: 0% (LMS, UPS), 11.1% (GIST), 14.3% (others)	Anemia (7.0%), fatigue (3.5%), lymphopenia (3.5%), oral mucositis (3.5%)
Gordon et al. (phase I/II) (SAINT) [94]	Ipi/nivo + trabectedin (trab)	Advanced STS	79	NA	ORR 25.3%; DCR 87.3% (6 CR *, 14 PR, 49 SD) mPFS 6.7 m; mOS 24.6 m * One surgical CR	ALT increase (25%), fatigue (8.7%), AST increase (8.7%), decreased neutrophil count (5.4%), anemia (4.6%)
Pink et al. (phase II) (NITRA-SARC) [95]	Nivo + trab	Advanced STS	25	NA	ORR 8%; DCR 48% (2 PR, 10 SD); mPFS 4 m	Leukopenia (47.2%), neutropenia (41.7%), thrombopenia (33.3%), increased ALT (30.6%), anemia (27.8%)
Smrke et al. (phase I) [96]	Pembro + gemcitabine	LMS, UPS	13	40–67	LMS (11): DCR 73% (8 SD) UPS (2): DCR 100% (2 PR)	NA
Nathenson et al. (phase II) [97]	Pembro + eribulin	LMS cohort	19	48–80	ORR 5.3%; DCR 26.3% (1 PR, 5 SD); mPFS 11 w	Most commonly, neutropenia, anemia, weight loss, diarrhea, lipase/ALP increase
Wagner et al. (phase I/II) [98]	Avelumab + trab	LMS, LPS	23	NA	ORR 13%; DCR 56% (3 PR, 10 SD); mPFS 8.3 m	NA
Toulmonde et al. (phase Ib) [99]	Durva + trab	Advanced STS cohort	16	NA	ORR 7%; PFS at 6 m: 28.6%	NA
Immune checkpoint inhibitors (ICI) + tyrosine–kinase inhibitors/antiangiogenic drugs						
Martin-Broto et al. (phase I/II) (IMMU-NOSARC)	Nivo + suni-tinib	BS cohort (17 OST, 14 CS, 8 ES, 1 UPS) [100]	40	21–74	ORR 5%; DCR 60% (1 CR, 1 PR, 22 SD) mPFS 3.7 m; mOS 14.2 m	Neutropenia (10%), anemia (10%), AST/ALT increase (7.5%), fatigue (5%), oral mucositis (5%), hemorrhage (2.5%), dysphagia (2.5%), thrombopenia (2.5%), malaise (2.5%), thromboembolism (2.5%), pneumonitis (2.5%)
		STS cohort [101]	43	19–77	ORR 9.3%; DCR 69.3% (1 CR, 3 PR, 26 SD); mPFS 5.9 m; mOS not reached (follow up 6.1 m)	AST increase (11.8%), ALT increase (9.8%), neutropenia (9.8%), fatigue (5.9%), thrombopenia (3.9%), diarrhea (3.9%), renal failure (3.9%)
Wilky et al. (phase II) [102]	Pembro + axitinib	12 ASPS, 6 LMS, 5 UPS, 2 DDLPS, 8 others (2 BS)	33	27–62	ORR 25%; DCR 53.1% (8 PR, 9 SD); mPFS (all): 4.7 m; mOS (all): 18.7 m; mPFS (ASPS): 12.4 m; mPFS (others): 3.0 m.	Hypertension (15%), autoimmune toxic effects (15%), nausea (6%), ALT/AST increase (3%), oral mucositis (3%), diarrhea (3%), abdominal pain (3%), hemoptysis (3%), hyperlipidemia (3%)
Xie et al. (phase II) [103]	Camre-lizumab + apatinib	CT-refractory OST	43	11–43	ORR 20.1% (9 PR) mDR 6.2 m mPFS 6.2 m; mOS 11.3 m	Wound dehiscence (14%), ALP increase (9.3%), AST/ALT increase (9.3%), blood bilirubin increase (9.3%), hypertriglyceridemia (7.0%), anorexia (7.0%), weight loss (7.0%), pneumothorax (7.0%), platelet count decrease (4.7%), diarrhea (4.7%), PPS (4.7%), limb pain (4.7%), leukopenia (4.7%), rash (4.7%), oral mucositis (4.7%), hypertension (4.7%), toothache (4.7%), nausea (4.7%), non-cardiac chest pain (4.7%), hypothyroidism (2.3%), LDH increase (2.3%), proteinuria (2.3%), cough (2.3%), hemorrhage (2.3%), fatigue (2.3%), peripheral neuroinflammation (2.3%)
Kim et al. (phase II) [104]	Durva + pazopanib	Advanced STS	47	NA	ORR 28.3% (1 CR, 12 PR) mPFS 8.6 m	NA
Cousin et al. (phase II) [105]	Avelumab + regorafenib	Advanced STS	43	NA	ORR 9.3%; DCR 48.8% (4 PR, 17 SD); mDR 7.8 m; mPFS 1.8 m; mOS 15.1 m	PPS (12.2%), fatigue (10.2%), diarrhea (10.2%)
Kelly et al. (phase II) [106]	Pembro + epacadostat	Advanced STS	29	24–78	ORR 3%; DCR 48% (1 PR, 13 SD); mPFS 8 w; PFS at 24 w 27.9%; mOS NA	AST increase (10%), ALT increase (3%), anemia (3%), hypophosphatemia (3%), lipase increase (3%)
Schöffski et al. (phase Ib) [107]	Pembro + olaratumab	Advanced STS	28	NA	ORR 21.4%; DCR 53.5%; mDR 16.2 m; mPFS 2.7 m; mOS 14.8 m	NA
Immune checkpoint inhibitors (ICI) + other agents						
Kelly et al. (phase II) [108]	Pembro + T-VEC	Advanced STS	20	24–90	ORR 35%; DCR 70% (7 PR, 7 SD); mPFS 17.1 w	Pneumonitis (5%), fever (5%), anemia (5%)
Chawla et al. (phase II) [109]	Trabectedin + nivo + T-VEC	Advanced sarcoma	36	NA	ORR 8.3%; DCR 86.1% (3 PR, 27 SD); mPFS 5.5 m; mOS 9.0 m; OS at 6 m 73%	Anemia (33.3%), ALT increase (22.2%), fatigue (11.1%), thrombopenia (11.1%), neutropenia (11.1%)
D’Angelo et al. (phase I) [110]	Nivo + bempegal-desleukin	Advanced STS	84	13–80	ORR 10.4% (PR: 3/8 in AS, 1/4 in ASPS, 2/10 in UPS, 1/10 in LMS, 1/10 in CS); mDR 9.3 m	Anemia (10%), lipase increase (10%), amylase increase (7%), hypertension (7%), pain (8%), thromboembolic events (5%)
Somaiah et al. (phase I) [111]	LV305	STS (13 SS, 6 MRCL)	24	25–72	ORR 4.2%; DCR 62.5% (1 PR, 14 SD); mPFS 4.6 m; mOS 33 m	No G3/G4 adverse events
Rosenbaum et al. (phase I) [112]	Avelumab + DCC-3014	Advanced STS (7 LMS)	13	32–71	DCR 23% (3 SD); decreased circulating MDSCs in 5/7 patients (median 26.9%)	ALT/AST increase (31%), CK increase (23%), amylase/lipase increase (16%), anemia (8%), hypertension (8%)
Chawla et al. (phase I) [113]	Avelumab + SNK01	Advanced sarcoma	15	20–75	ORR 13.3%; DCR 33.3% (2 PR, 3 SD); mPFS 11.1 w	No G3/G3 adverse events related to SNK01
Therapeutic vaccines						
Kawaguchi et al. (phase I) [114]	SYT-SSX vaccine	SS	21	21–69	DCR 50% (6 SD out of 12 assessable patients)	NA
Takahashi et al. (phase II) [115]	Peptide vaccine	Advanced sarcoma	20	23–75	DCR 30% (6 SD); mOS 9.6 m	NA
Pipia et al. (phase I/II) [116]	DC vaccine	Advanced STS	74	NA	Cohort 1 (adj/maint): mOS 24.4 m; cohort 2 (mono): mOS 14.2 m	NA
Chawla et al. (phase II) [117]	Atezo +/− CMB305	SS, MLS	89	NA	mPFS 2.6 m vs. 1.6 m mOS 18 m in both groups	4 G3/G4 events in each group (not specified)
Adoptive cell therapy						
Robbins et al. (phase II) [118]	NY-ESO1 TCR	SS	18	19–65	ORR 61% (1 CR, 10 PR); PFS 3–47 m; estimated 3-y OS: 38%	No toxicities attributed to the transferred cells *
D’Angelo et al. (phase II) [119]	NY-ESO1 TCR	SS	45	NA	ORR 33% (1 CR, 14 PR); mPFS 8.6–22.4 w	No toxicities attributed to the transferred cells *
Van Tine et al. (phase I) [120]	MAGE-A4 TCR	SS	8	NA	ORR 50%; DCR 87.5% (4 PR, 3 SD)	No toxicities attributed to the transferred cells *
D’Angelo et al. (phase II) [121]	MAGE-A4 TCR	STS (23 SS,2 MLS)	25	24–73	ORR 40%; DCR 84% (2 CR, 8 PR, 11 SD)	CRS (5%) *
Ahmed et al. (phase I/II) [122]	Her2-CAR T cells	Advanced sarcoma (16 OST, 3 others)	19	7–29	DCR 23.5% (4 SD); mOS 10.3 m	Anemia (5.3%), muscle weakness (5.3%), back pain (5.3%)
Navai et al. (phase I) [123]	Her2-CAR T cells	Advanced sarcoma (5 OST, 5 others)	10	4–54	ORR 20%; DCR 50% (2 CR, 3 SD) (CR in 1 OST and 1 RMS)	No toxicities attributed to the transferred cells *

* All patients experienced transient neutropenia and thrombopenia induced by the lymphodepleting CT with fludarabine plus cyclophosphamide and the transient toxicities associated with IL-2 infusion.

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
