# Peer review of "Current Landscape of Immunotherapy for Advanced Sarcoma"

_cancers, 2023, doi:10.3390/cancers15082287_

Round 1

Reviewer 1 Report

First, I would like to to congratulate the authors to this truly well written review regarding an important therapeutic approach in the sarcoma field.

However, the review contains mainly data and publications regarding Check-point inhibition and small chapters regarding other immune therapy approaches, such as adoptive T- Cell transfer.

Totally unmentionend is the approach of antibody drug conjugates, with numerous studies in place right now, e.g. NCT02487979, NCT02565758, NCT02452554, NCT04650984 (FIBROSARC), NCT04733183 (FLASH). Also combination studies are not completely listed, such as NCT03590210 (NITRASARC). Finally I would suggest to list at least GD2 as target in immune therapy for sarcoma, which is used in adoptive T-Cell studies and for conjugates (e.g. Journal of Pediatric Hematology/Oncology 44(6):p e948-e953, August 2022).

This would really complete the picture and better reflect the chosen title of the review: "presence and FUTURE".

Author Response

Thank you so much for your comments.

The results of NITRASARC trial have been included in the manuscript as suggested (lines 469-471). We submit a revised version of the article.

GD2 has already been mentioned in our manuscript as a target for adoptive cell therapy (lines 695-696). We have prioritized the detailed explanation of those targets whose drugs have achieved further clinical development.

For this same reason, the studies evaluating immune checkpoint inhibitors have been explained in detail, since most of available clinical trials have used these drugs, alone or in combination with other agents. Considering that the purpose of this article is to summarize the clinical (and not basic) research on immunotherapy for sarcomas, other strategies with a scarcer clinical development (such as adoptive cell therapy) have been mentioned more succinctly on purpose.

Regarding antibody-drug conjugates, most of these molecules target specific proteins which are over-expressed on the cell surface, incorporating a cytotoxic agent which is selectively delivered to the tumor environment. We think this is a completely different therapeutic approach and should not be considered as an immune treatment. This is the reason why the studies evaluating ADCs in sarcoma have not been included in the manuscript, which is focused on immunotherapy strategies. We sincerely think that an appropriate approach to the use of ADCs should be the purpose of a different review and is out of the scope of this article.

Reviewer 2 Report

In the review manuscript entitled "Immunotherapy in sarcoma: present and future", Albarran et al. concisely summarized the basis of immunotherapy both in general and in sarcomas and the results of clinical trial of immunotherapy for sarcomas. A concern of this reviewer is that the manuscript is merely a summary of other papers and there are almost no opinions or comments from the authors as experts in this field.

Comments:

1.      As I mentioned above, there are no opinions or comments throughout the manuscript. For example, even the Conclusion session is a repeated summary of the basis of immunotherapy and clinical trials, and it does not state about the "future" of this field unfortunately (therefore the title of this manuscript is not appropriate).

2.      The differences in the response rates of combined ICIs and CT are huge among these studies in page 14. The authors could comment on the potential reasons.  

3.      How the T-VEC was engineered to selectively replicate within cancer cells? The authors should provide more detailed explanation.

4.      There are many typos. For example, it should be synovial "sarcoma(s)" in line 90. Lines 108-127 are duplicate (it may be an error in my PDF viewer).

5.      The first appearance of "DCR" is in line 385 and not in line 400. ")" is missing in line 652.

6.      Please don't use "OS" for osteosarcoma. Overall survival is also represented as OS in the later section and they are confusing.

Author Response

Thank you so much for your comments.

An alternative title might be: “current landscape of immunotherapy for advanced sarcoma”, which may fit better with the content of the review.

Several comments and subjective opinions have been included throughout the manuscript (though the main purpose of the article is to summarize the current evidence and clinical studies, and that is why a significant part of the text is dedicated to this in detail).

We have added a new paragraph explaining the potential reasons why there are significant differences in the results of the ICI/CT combination trials (lines 482-486).

The mechanism of action of T-VEC has been explained more clearly (lines 552-555).

“Sarcoma” has been added in line 90. The duplicated paragraph has been deleted.

The mistake in “DCR” abbreviation has been corrected.

“OS” has been replaced by “osteosarcoma” when needed.

Round 2

Reviewer 2 Report

It is a great dissapointment that the authors ignored my first and most important comment as copied below. Given that there are so many similar review papers in the literature, experts' comments and opinions would be almost the only way to find any novelty in this manuscript. 

In the review manuscript entitled "Immunotherapy in sarcoma: present and future", Albarran et al. concisely summarized the basis of immunotherapy both in general and in sarcomas and the results of clinical trial of immunotherapy for sarcomas. A concern of this reviewer is that the manuscript is merely a summary of other papers and there are almost no opinions or comments from the authors as experts in this field.

Comments:

1.      As I mentioned above, there are no opinions or comments throughout the manuscript. For example, even the Conclusion session is a repeated summary of the basis of immunotherapy and clinical trials, and it does not state about the "future" of this field unfortunately (therefore the title of this manuscript is not appropriate).

Author Response

Dear reviewer, we have submitted a corrected version of the manuscript, including some new paragraphs that incorporate our point of view about the role of these therapies. Regarding the use of ICI as monotherapy, see lines 446-456 (page 13). About the combination of ICI plus chemotherapy, see lines 519-528 (page 14). Regarding the use of ICI plus TKI, see lines 585-593 (pages 15-16). Thank you again for your feedback.